# Bayesian workflow for time-varying transmission in stratified compartmental infectious disease transmission models

**Judith A. Bouman**[1,2]*, **Anthony Hauser**[1,3], **Simon L. Grimm**[1,4], **Martin Wohlfender**[1,2], **Samir Bhatt**[5,6], **Elizaveta Semenova**[7], **Andrew Gelman**[8,9], **Christian L. Althaus**[1,2☯], **Julien Riou**[10☯]*

**1** Institute of Social and Preventive Medicine, University of Bern, Bern, Switzerland, **2** Multidisciplinary Center for Infectious Diseases, University of Bern, Bern, Switzerland, **3** Institut national de la santé et de la recherche médicale Sorbonne Université (INSERM), Sorbonne Université, Paris, France, **4** Center for Space and Habitability, University of Bern, Bern, Switzerland, **5** MRC Centre for Global Infectious Disease Analysis, Jameel Institute, School of Public Health, Imperial College London, London, United Kingdom, **6** Section of Epidemiology, Department of Public Health, University of Copenhagen, Copenhagen, Denmark, **7** Department of Epidemiology and Biostatistics, Imperial College London, London, United Kingdom, **8** Department of Statistics, Columbia University, New York, New York, United States of America, **9** Department of Political Science, Columbia University, New York, New York, United States of America, **10** Department of Epidemiology and Health Systems, Unisanté, Center for Primary Care and Public Health & University of Lausanne, Lausanne, Switzerland

☯ These authors contributed equally to this work.
* judith.bouman@unibe.ch (JAB); julien.riou@unisante.ch (JR)

**Data Availability Statement:** Code and data are publicly available as an R-package (https://github.com/JudithBouman2412/HETTMO).

## Abstract

Compartmental models that describe infectious disease transmission across subpopulations are central for assessing the impact of non-pharmaceutical interventions, behavioral changes and seasonal effects on the spread of respiratory infections. We present a Bayesian workflow for such models, including four features: (1) an adjustment for incomplete case ascertainment, (2) an adequate sampling distribution of laboratory-confirmed cases, (3) a flexible, time-varying transmission rate, and (4) a stratification by age group. Within the workflow, we benchmarked the performance of various implementations of two of these features (2 and 3). For the second feature, we used SARS-CoV-2 data from the canton of Geneva (Switzerland) and found that a quasi-Poisson distribution is the most suitable sampling distribution for describing the overdispersion in the observed laboratory-confirmed cases. For the third feature, we implemented three methods: Brownian motion, B-splines, and approximate Gaussian processes (aGP). We compared their performance in terms of the number of effective samples per second, and the error and sharpness in estimating the time-varying transmission rate over a selection of ordinary differential equation solvers and tuning parameters, using simulated seroprevalence and laboratory-confirmed case data. Even though all methods could recover the time-varying dynamics in the transmission rate accurately, we found that B-splines perform up to four and ten times faster than Brownian motion and aGPs, respectively. We validated the B-spline model with simulated age-stratified data. We applied this model to 2020 laboratory-confirmed SARS-CoV-2 cases and two seroprevalence studies from the canton of Geneva. This resulted in detailed estimates of the transmission rate over time and the case ascertainment. Our results illustrate the

**Funding:** This study is funded by the Multidisciplinary Center for Infectious Diseases, University of Bern, Bern, Switzerland. JR is supported by the Swiss Federal Office of Public Health (142005806) and by the Swiss National Science Foundation (189498). CLA received funding from the European Union's Horizon 2020 research and innovation program - project EpiPose (No 101003688) and the Swiss National Science Foundation (No 196046). S.B. acknowledges support from the MRC Centre for Global Infectious Disease Analysis (MR/R015600/1), jointly funded by the UK Medical Research Council (MRC) and the UK Foreign, Commonwealth & Development Office (FCDO), under the MRC/FCDO Concordat agreement, and also part of the EDCTP2 programme supported by the European Union. S.B. acknowledges support from the National Institute for Health and Care Research (NIHR) via the Health Protection Research Unit in Modelling and Health Economics, which is a partnership between the UK Health Security Agency (UKHSA), Imperial College London, and the London School of Hygiene &; Tropical Medicine (grant code NIHR200908). (The views expressed are those of the authors and not necessarily those of the UK Department of Health and Social Care, NIHR, or UKHSA.). S.B. acknowledges support from the Novo Nordisk Foundation via The Novo Nordisk Young Investigator Award (NNF20OC0059309). SB acknowledges the Danish National Research Foundation (DNRF160) through the chair grant. S. B. acknowledges support from The Eric and Wendy Schmidt Fund For Strategic Innovation via the Schmidt Polymath Award (G-22-63345). E.S. acknowledges support in part by the AI2050 program at Schmidt Futures (Grant [G-22-64476]). A.G. acknowledges funding from the Office of Naval Research. The funders had no role in study design, data collection and analysis, decision to publish, or preparation of the manuscript.

**Competing interests:** The authors have declared that no competing interests exist.

potential of the presented workflow including stratified transmission to estimate age-specific epidemiological parameters. The workflow is freely available in the R package HETTMO, and can be easily adapted and applied to other infectious diseases.

## Author summary

Mathematical models are a central tool for understanding the spread of infectious diseases. These models can frequently be fitted to surveillance data such as the number of laboratory-confirmed cases and seroprevalence over time. We identified that in these situations, four crucial features are required for a model to provide insightful information for managing an epidemic. These features relate to the adjustment for incomplete case ascertainment, to the choice of sampling distribution, to the variation of transmission over time and to the stratification by age. For each feature, we identify and compare several implementation options on simulated data. This structural comparison of methods results in a Bayesian workflow that is optimized for modeling the transmission of SARS-CoV-2 over a short period. We highlight the advantages and limitations of our approach in a real situation, using real-world SARS-CoV-2 data from the canton of Geneva. In addition to providing validated solutions to important technical points, such a comprehensive workflow helps to improve the reliability and the transparency of epidemic models.

## Introduction

Epidemic theory provides mathematical expressions for biological concepts that are fundamental to understanding the spread of infectious diseases, such as contagion, incubation and immunity. Compartmental models based on ordinary differential equations (ODEs) implement these concepts within a unified, manageable framework, and have taken a central position in the field of infectious disease modeling. While initially used to formalize and develop theoretical notions such as reproductive numbers or immunity thresholds [1], or to simulate epidemics under specific constraints [2, 3], compartmental transmission models have been increasingly applied to practical questions about infectious disease transmission, especially during the COVID-19 pandemic [4–6]. These applications often rely upon fitting custom-made models to surveillance data such as counts of laboratory-confirmed cases, and use various methods of statistical inference. Among these, Bayesian inference with Markov chain Monte Carlo (MCMC) is gaining ground, fueled by improvements in computing power and sampling algorithms [7], and by efficient software implementations [8–10]. This approach offers many advantages, including parameter inference, full propagation of uncertainty, principled integration of prior knowledge and high flexibility in model specification [11]. Still, even the most basic situations require models of relatively high complexity, with many options available for each model feature, and difficulties of implementation and computational inefficiency limit the widespread adoption of these tools. In such situations, it is beneficial to describe the entire iterative process of model development, evaluation, and refinement, rather than presenting a single Bayesian model, since the optimal model choices frequently depend on the specifics of the real-world scenario. Gelman et al. (2020) have defined this comprehensive process as the Bayesian workflow, which utilizes simulated data to verify the accuracy of the inference and compare model efficiency [12].

We identified four essential features for a Bayesian workflow aimed at studying the transmission of SARS-CoV-2 (or other respiratory viruses) in a population over a relatively short time period based on counts of laboratory-confirmed cases: (1) an adjustment for incomplete case ascertainment, (2) an adequate sampling distribution of laboratory-confirmed cases, (3) a flexible, time-varying transmission rate, and (4) a stratification by age group. First, incomplete and unrepresentative ascertainment plays a key role in the generation of surveillance data. Indeed, laboratory-confirmed cases are only an unrepresentative subset of the actual population of newly infected individuals, that is highly dependent on testing activity (how many tests are performed) and targeting (which part of the population is prioritized for or has access to testing), both of which can vary over time [13–15]. The identification of the ascertainment rate, however, requires additional information such as point estimates of population seroprevalence [11]. Second, the sampling distribution must be suitable to generate counts of laboratory-confirmed cases. Common options include Poisson, quasi-Poisson and negative binomial distributions, but no systematic comparison in this context has been conducted to date [16, 17]. The third feature, flexible time-varying transmission, is critical, as it models the variations in transmission caused by drivers such as non-pharmaceutical interventions (NPIs), alterations of behaviors, and environmental determinants. These drivers can impact both components of the transmission rate: the rate of contact between individuals (e.g., mandatory work from home) and the probability of transmission upon contact (e.g., mandatory face masks). As these factors may vary over time, any model aimed at disentangling and understanding the drivers of SARS-CoV-2 transmission must incorporate a time-varying transmission rate. Several approaches have been proposed using predefined functional shapes [18–21] or more flexible approaches based on step functions [5], cubic splines [22–24] or Brownian motion [25–27]. A systematic comparison of these methods in the context of compartmental transmission models is currently lacking. Fourth, the stratification by age group is now considered standard practice in transmission models of respiratory infections [28]. Indeed, age influences every step of the infection course of SARS-CoV-2 and other respiratory viruses including contact patterns, adherence to NPIs, probability of testing and probability of severe outcome [27, 29–31]. While other individual factors like gender and socio-economic position may certainly influence transmission [32], age is generally considered as the most important, justifying this first choice for stratification.

In this work, we present a Bayesian workflow for a compartmental transmission model to analyze the transmission of SARS-CoV-2 that includes these four essential features. To this aim, we assess the statistical accuracy and computational efficiency of several of these model choices, including three sampling distributions (Poisson, quasi-Poisson, and negative binomial) and three methods for implementing a time-varying transmission rate (Brownian motion, B-splines, and approximate Gaussian processes). For these assessments, we use both simulated data and real-world data from SARS-CoV-2 in Geneva, Switzerland. We release the scripts and functions for the different model iterations presented in this study in an R package called HETTMO (for HETerogeneous Transmission MOdel). The code can easily be adapted to other situations and pathogens, with the objective of promoting and facilitating access to this type of methods and making the process of model validation and comparison insightful.

## Materials and methods

We preregistered our methodology for this study on the Open Science Framework (OSF). This pre-registration document can be accessed at https://osf.io/n73gu/?view_only=4e469db4a58d428f99682e38c81f0d58.

## Transmission model

At its core, the compartmental infectious disease transmission model follows a Susceptible-Exposed-Infected-Removed (SEIR) structure (Fig 1). We extended this model by allowing the transmission rate to vary over time. The model definition is shown in Eq 1, where $\rho(t)$ is the time-dependent factor that describes the change in the transmission rate over time relative to a baseline. The probability of a transmission event upon contact is $\beta$, $\tau$ is the inverse of the average latent period, $\gamma$ the inverse of the average time an individual spends in the infected compartment (i.e. the recovery rate) and $c$ is the average number of contacts an individual has per day. The total population is $N = S + E + I + R$. Both $\tau$ and $\gamma$ are fixed, such that the generation time is 5.2 days, with this time equally distributed between the exposed and infected compartments [33–35]. We have chosen to parameterize $\gamma$ and $\tau$ together based on the generation time rather than with independent estimates for the latent period and the recovery rate. The reason is that we assume that individuals will isolate after a positive PCR test during the study period and therefore move to the R compartment earlier than would be expected based on the recovery rate. Moreover, since our study period was less than a year, we could assume that there is no waning of immunity and that the total population size is constant.

$$\frac{dS}{dt} = -\beta\rho(t)cS\frac{I}{N}$$
$$\frac{dE}{dt} = \beta\rho(t)cS\frac{I}{N} - \tau E$$
$$\frac{dI}{dt} = \tau E - \gamma I \tag{1}$$
$$\frac{dR}{dt} = \gamma I$$

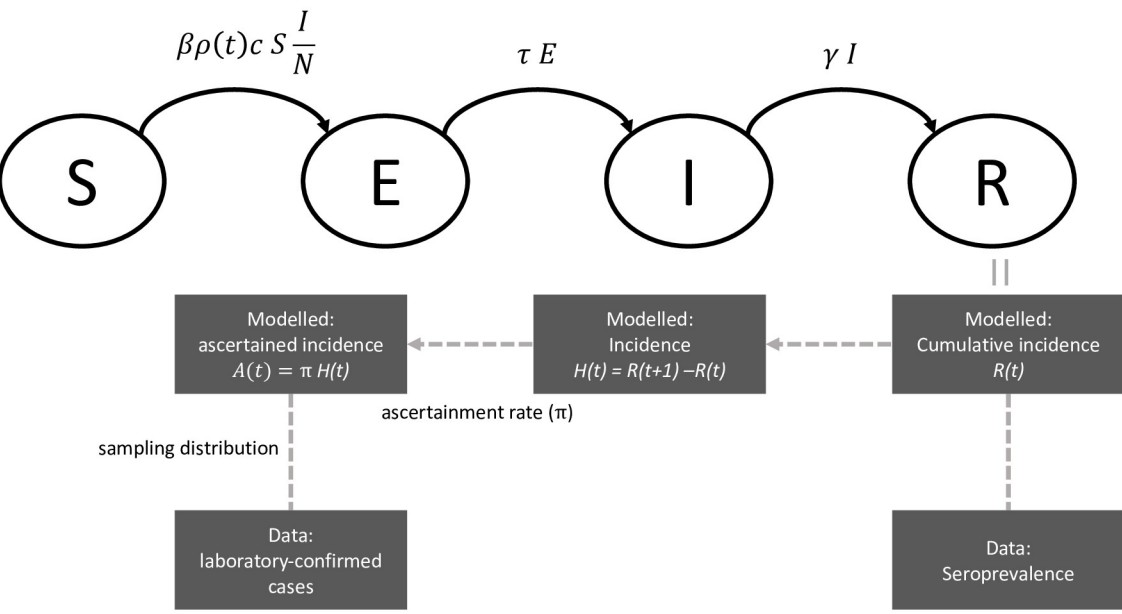

**Fig 1. Schematic overview of the SEIR transmission model for SARS-CoV-2 and the steps to generate the number of laboratory-confirmed cases and the observed seroprevalence.**

**Feature 1: Adjustment for incomplete case ascertainment using seroprevalence data**
From a given set of parameter values and initial conditions, the SEIR model generates the total number of newly recovered individuals in the population by unit of time (i.e. the true incidence by our definition, see Fig 1) as follows:

$$H(t) = R(t+1) - R(t).$$

Only a fraction of this incidence will be ascertained as a laboratory-confirmed cases by testing positive ($A(t)$). The ascertainment rate $\pi$ determines what fraction of the true incidence is observed: $A(t) = \pi H(t)$. It is influenced by many determinants including testing activity and targeting, and may thus also vary over time. In this context the ascertainment rate is not statistically identifiable without the support of external data, such as a seroprevalence estimate. The seroprevalence is a measure of the number of recovered individuals in the population at a given time (if antibody waning and vaccination can be ignored), and thus informs about the cumulative true incidence over a period of time. Assuming that ascertainment is stable for that period of time, seroprevalence data can be used to estimate the ascertainment rate and anchor the model. In practice, we assume that the SEIR model also generates the cumulative number of removed individuals in the population at time $t$ (from the $R$ compartment), which is linked to seroprevalence data at time $t$ using a simple binomial sampling distribution. We thus define periods bounded by seroprevalence studies, and estimate one ascertainment for each period. We also correct the seroprevalence data for imperfect testing [13].

**Feature 2: Sampling distribution for weekly laboratory-confirmed cases** We account for process noise in the transmission and observational noise in the ascertainment of cases by introducing a sampling distribution generating counts of laboratory-confirmed cases given the ascertained incidence. Process noise results from overdispersion of cases due to stochastic processes that are not captured by the compartmental transmission model, and observational noise from sampling of cases. Here, we compare several options based on data from the Swiss canton of Geneva in 2020. First, we try a Poisson distribution, where the variance is equal to the mean $\lambda$. We then consider two distributions that include an additional overdispersion parameter $\theta$: a quasi-Poisson model, where the variance is a linear function of the mean ($\theta\lambda$) and a negative binomial distribution, where the variance is a quadratic function of the mean ($\lambda + \lambda^2\theta$) [16].

**Feature 3: Flexible, time-varying transmission** In the compartmental transmission model, time-variation in transmission is controlled by the forcing function $\rho(t)$, which applies to the contact rate $c$ and the probability of transmission upon contact $\beta$ at the same time. Therefore, time variation in these two components is considered together, and it is not possible to disentangle between them. We compare the performance and efficiency of three different methods to implement the time-varying transmission: Brownian motion, B-splines, and approximate Gaussian processes:

1. We implemented Brownian motion as a Gaussian random walk similar to Bouranis et al (2022) with weekly time-steps; see Eqs 2 and 3, taken from Bouranis et al (2022) [27]. In these equations, $t$ is the discrete weekly time step, and $W$ a random process whose elements are normally distributed with mean 0 and variance $s$. This value $s$ is estimated from the data given a normal prior. This approach creates prior functions for $\rho(t)$ with increasing variance over time [36]:

$$\eta(t+1) = \eta(t) + W_t \text{ and } \eta(0) = W_0, \tag{2}$$

$$\rho(t) = \exp \eta(t) \tag{3}$$

2. Our implementation of the B-splines relies on the functions provided by Kharratzadeh [37]. B-splines are uniquely defined by the degree of the polynomials and the predefined set of knots. To be able to use the splines within the ODE system, without recomputing them every time the ODE is evaluated (that is, multiple times per MCMC iteration), we calculated the value of the B-splines for degree-1 points between two consecutive knots. Based on these values, we calculated the coefficients of a polynomial based on the degree of the B-spline using the Lagrange algorithm. These coefficients are then used as an input variable for the ODE model. For each iteration in the MCMC, a set of coefficients is sampled that defines how the B-splines must be combined to create the transmission rate function over time. In addition, this approach requires setting values for the knots. We consider five different sets of knots (Table 1) all in combination with cubic splines.

3. Gaussian processes (GPs) are powerful and flexible fitting tools for modeling time series that are increasingly used in the field [38, 39]. We use a Gaussian process with an exponentiated quadratic covariance function, which, to our knowledge, has not yet been applied to compartmental transmission models. To reduce the computational cost, our implementation follows the proposition of Riutort-Mayol et al. (2020), using a basis function approximation via Laplace eigenfunctions, itself based on the mathematical theory developed by Solin and Särkkä (2020) [40, 41]. This low-rank Bayesian approximation requires several tuning parameters, most importantly the number of basis functions $M$ and the boundary factor $c$, that determines the interval at which the approximation of the GP is valid. This interval is then given by the range of values at which the data is observed multiplied with the boundary factor. Both $M$ and $c$ influence the accuracy and the efficiency of the algorithm [40]. We test and compare the performance of the algorithm for a set of boundary factors and increasing number of basis-functions to find optimal values for the type of function we expect in our epidemiological data. Besides the number of basis functions and the boundary factor, the GP approximation also requires a parameter for the length scale ($L$, controlling the sinuosity of the basis-functions) and the marginal variance ($A$). As the length scale and the marginal variance both influence the smoothness of the function, they are unidentifiable in our set-up. We therefore fix the marginal variance to 0.5 and estimate the length scale from the data.

Both the Brownian motion and the B-splines are special cases of a Gaussian process given a specific kernel. However, our implementation of these methods differs from the implementation of the approximate Gaussian processes (aGPs).

**Feature 4: Stratification by age group** We consider three age groups in order to limit the computational cost: 0–19 years old, 20–64 years old and 65 and older. The stratification is

**Table 1. Knot sequences.**

| Knot sequence identifier | Location of first knot | Period between knots |
| --- | --- | --- |
| 1: true knots | 4 weeks | 4 weeks |
| 2: 8 weeks | 8 weeks | 8 weeks |
| 3: 12 weeks | 12 weeks | 12 weeks |
| 4: 4 weeks shifted | 6 weeks | 4 weeks |
| 5: 8 weeks shifted | 6 weeks | 8 weeks |

Overview of different sequences of knots used for the B-spline method to analyze time-dependent transmission rates in an compartmental infectious disease transmission model.

implemented by replacing the contact rate $c$ with a 3x3 contact matrix that indicates the average number of contacts an index case of a given age group (in the column) has with individuals of the other age groups (rows). We use a synthetic contact matrix as our pre-COVID baseline, as empirical data for Switzerland are lacking for this time period (S4 Table). For this, we rely on the work of Prem et al. (2021) and rescale their suggested social contact matrix for Switzerland to match the age-distribution in our defined age groups in the canton of Geneva [42]. Stratification also applies to the processes of ascertainment, time-varying transmission and sampling that now occur independently by age group, hereby multiplying the number of parameters to estimate by three. The equation in S1 Text shows the ODE system for the stratified version of the SEIR transmission model.

## Bayesian inference

We consider the models in a Bayesian framework, with the objective of estimating $\beta$, $\rho(t)$, $\pi_t$, and where relevant, $\theta$, from two data sources: weekly counts of laboratory-confirmed cases of SARS-CoV-2 infection (this would also apply to any other respiratory virus) and one or more seroprevalence estimates. When relevant, these data need to be stratified by age group in the same way. We use weakly informative prior distributions for these parameters (S1 Text). The different versions of the model are implemented in Stan, a platform for Bayesian inference [8, 43]. Stan allows for coding a large variety of model features, relying on a few principles to optimize computational efficiency. For a detailed description of how to implement and scale-up ODE-based models, see Grinztajn et al. (2021) [11]. A key aspect here is the choice of the numerical ODE solver. To continue with our objective of identifying the most optimal implementation of the model in this type of situation, we compared all forward sensitivity solvers currently available in Stan: "rk45" (4th and 5th order Runge–Kutta-Fehlberg [44, 45]), "adams" (Adams-Moulton formula [46, 47]), "bdf" (backward differentiation formula [46, 47] and "ckrk" (fourth and fifth order explicit Runge-Kutta method for non-stiff and semi-stiff systems [45, 48]). We also compare to a simple solver that uses the trapezoidal rule to approximate the solution of the system as described in Bouranis et al [27]. For the trapezoidal solver, we use twenty equidistant time steps within each (weekly) time step in the model. Besides the solver itself, we also test different tuning values for the solver tolerance (1e-4, 1e-5 and 1e-6; not relevant for the trapezoidal solver) and the number of warm-up-iterations (300 and 500). All combinations are run for 8 chains with 250 MCMC sampling-iterations.

## Simulated data

We validate and compare the different versions of the model with simulated data of an epidemic of a respiratory pathogen. The simulation study is conducted in two steps. In the first step, we assume that the population is well-mixed, and ignore the age stratification. We simulate data of laboratory-confirmed cases for 45 weeks and 100, 000 individuals, with two successive epidemic waves. We also simulate seroprevalence data after 20 weeks and at the end of the simulation, thus defining two periods with ascertainment $\pi_1$ and $\pi_2$, respectively. The transmission process is modeled with a probability of transmission per contact of $\beta = 8.5\%$, a baseline of $c = 11$ contacts per day [42] and a time-varying component based on a spline of degree 3 and a knot every 4 weeks. We set ascertainment at $\pi_1 = 0.3$ and $\pi_2 = 0.5$. S1 Table provides an overview of all parameter values chosen to create the simulated data. We generate one simulated dataset and apply all model versions to these data. We evaluate predictive performance by computing the root mean squared error (RMSE) between the estimated and true value of $\rho(t)$, an RMSE weighted by the number of laboratory-confirmed cases per week, and evaluate

computational performance by comparing the number of effective samples per second. In a second step, we select the best performing model version from step 1, add the stratification by age, and validate again on stratified simulated data. We modify several parameters to simulate a stratified dataset with three age classes (S1 Table). The simulated data are available in the HETTMO R package.

### Data from the canton of Geneva in 2020

Finally, we apply the best performing model version to weekly counts of laboratory-confirmed cases of SARS-CoV-2 infection in the canton of Geneva in 2020 (data from the Federal Office of Public Health). During this time period, we could reasonably assume that there was no waning of immunity and that vaccination did not yet influence transmission dynamics. Moreover, in Geneva, two seroprevalence surveys were performed during this time period: the first one from April 6th until May 10th [13], and the second from November 23th until December 23st [49]. The results are summarized in S3 Table. Both surveys use the EuroImmune IgG test (Euroimmun; Lübeck, Germany #EI 2606–9601 G), which has a sensitivity of 93% and a specificity of 100% for the cutoff suggested by the manufacturer [50]. We aggregate all data according to the age groups used in the first serosurvey (0–19 years old, 20–64 years old and 65 and older). As the second serosurvey uses a different grouping, we reallocate the results in age group 18–24 to age groups 0–19 and 20–59 using the age-distribution in the population of the canton of Geneva as recorded by the Federal Office of Statistics [51]. These data are available in the HETTMO R-package.

### Data from the canton of Vaud

For an additional example of our approach, we apply the B-spline based model to data from the canton of Vaud, Switzerland. The laboratory-confirmed SARS-CoV-2 cases are obtained from the Federal Office of Public Health. The serosurvey was performed by the Corona Immunitas Research Group in Switzerland [52]. As the seroprevalence data were collected over a prolonged period, we decided to use monthly seroprevalence estimates for fitting the model. We assume two distinct ascertainment rates, one for the start of the epidemic until the 27th of July 2020 and one for after the 27th of July.

### Software implementation

We use R version 4.2.1 [53]. We published a R package called HETTMO that contains all functions needed to perform the analysis and run the models. HETTMO is the acronym for "heterogeneous transmission model" since the package can be used to model a heterogenous population that is stratified by, for example, age. The package is based on Stan (version 2.21.7) [43] and the cmdstanr package (version 0.5.3) [54]. HETTMO is available on GitHub at https://github.com/JudithBouman2412/HETTMO. Calculations for Figs 2 and S1–S4 were performed on UBELIX (https://www.id.unibe.ch/hpc), the HPC cluster at the University of Bern.

### Generality of the proposed approach

The HETTMO package includes the scripts and functions for the different model iterations presented in this study, which form the Bayesian workflow together with the model comparison. This workflow is tailored to study the early spread of SARS-CoV-2, justifying the choice of an SEIR model, the different assumptions on how model outputs are linked to the data, and the specific parametrization. Other situations and pathogens would likely require adaptations,

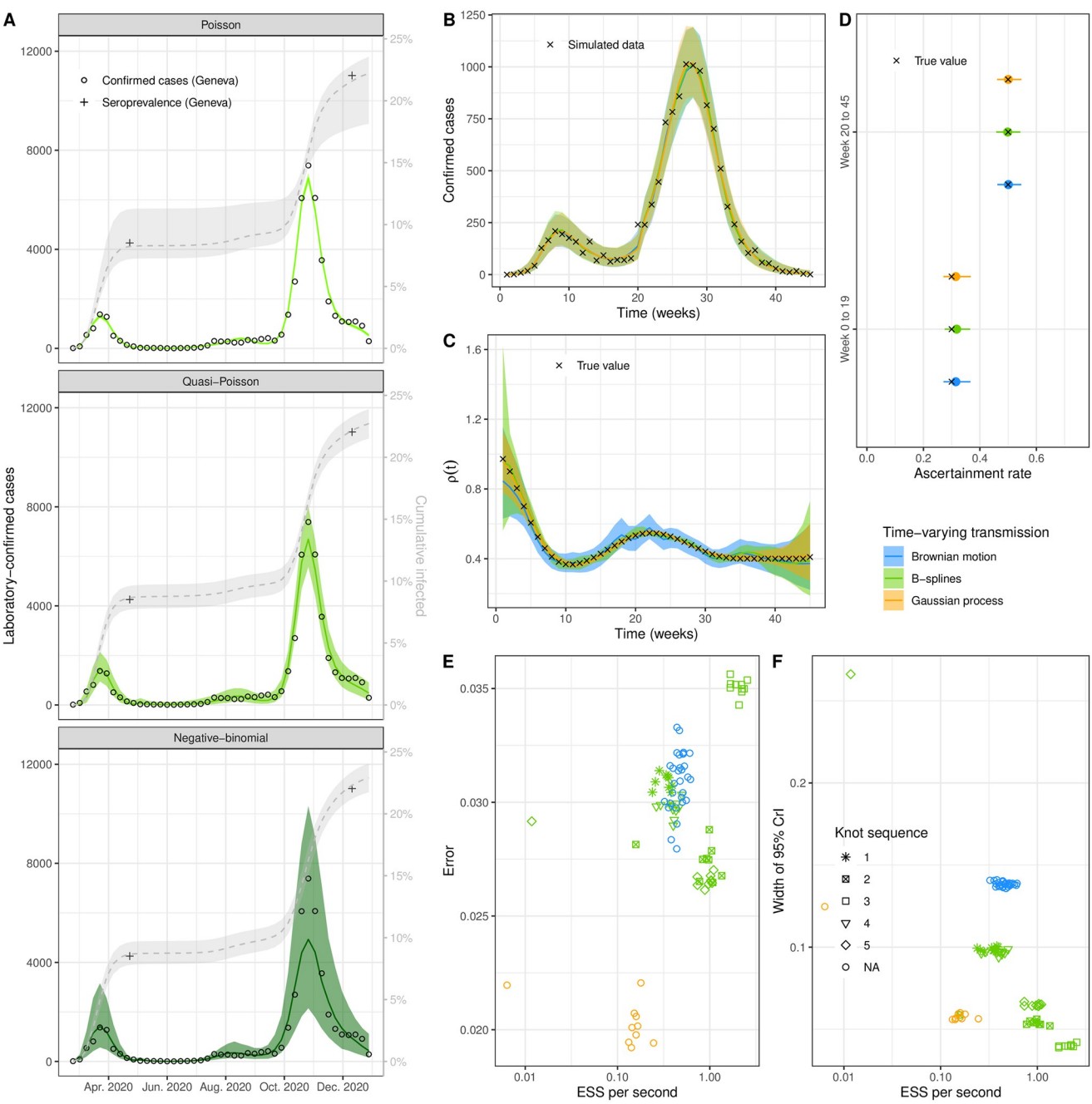

**Fig 2. Result from unstratified models.** (A) Posterior predictive plot for laboratory-confirmed cases (left y-axis, green ribbon) and cumulative incidence (right y-axis, gray ribbon) of SARS-CoV-2 in the canton of Geneva, Switzerland, for three iterations of the model with different sampling distributions (Poisson, quasi-Poisson and negative-binomial). Circles are weekly counts of laboratory-confirmed cases and pluses are estimates of seroprevalence at two time points. (B-D) Comparison of three methods of implementation of time-varying transmission on simulated data of a SARS-CoV-2 epidemic (posterior predictive plot, time-varying transmission $\rho(t)$, and ascertainment rate by period). (E-F) Benchmark of different implementations of time-varying transmission on simulated data of a SARS-CoV-2 epidemic, with performance expressed in effective sample size (ESS) per second, error defined as the difference between the median posterior and true $\rho(t)$, and the width of the 95% credible interval of $\rho(t)$ as a measure for precision. See Table 1 for details about the knot sequence.

such as additional compartments for vaccinated individuals or waning of immunity. Nevertheless, the presented case-study remains relevant in the wider context of infectious disease modeling. The four crucial features that were identified are broadly applicable for modeling many infectious diseases, and our comparisons regarding the most efficient implementations likely apply to other situations. In addition, our workflow can be directly used as a basis for other models, both technically, by providing easily adaptable code, and conceptually, by showcasing the advantages of the Bayesian workflow in this context.

## Results

We present a Bayesian workflow to find an optimal compartmental transmission model aimed at analyzing the transmission of a respiratory virus, with SARS-CoV-2 as a case-study, in a population over a time period short enough so that immunity waning can be ignored (a few months or years). We focus on four aspects, representing four features deemed as essential in this situation. First, we validate in a simulation study that our models, jointly fitted to both laboratory-confirmed cases and seroprevalence data, are able to provide accurate and unbiased estimates of the ascertainment rate by periods of time bounded by serosurvey estimates (feature 1). We find that the appropriate handling of uncertainty in these models is largely influenced by the choice of sampling distributions (feature 2). We investigate the most adequate sampling distributions for laboratory-confirmed cases of SARS-CoV-2 using real-world data from the canton of Geneva (Switzerland). Whereas the Poisson and negative-binomial distribution under- and overestimates the variability in laboratory-confirmed cases, respectively, we found that the quasi-Poisson distribution, with the variance scaling linearly with the mean, better fits the variability of the data (Fig 2A).

The next step in the Bayesian workflow is to benchmark several implementations of the time-varying transmission in a simulation study (feature 3). These different approaches all use flexible parameterizations of forcing functions, estimated from data. In a systematic comparison, we confirm that implementations based on Brownian motion, B-splines, and aGPs lead to very similar model fits (Fig 2B). The estimation of the variation in transmission over time $\rho(t)$ is accurate and unbiased under all three approaches (Fig 2C), but the Brownian motion approach overestimates the uncertainty. The estimation of the ascertainment rate $\pi_t$ is accurate under all three approaches, with a small overestimation in the first epidemic wave (Fig 2D).

Benchmarking the tuning parameters (the type of ODE solver, tolerance of the solver, and number of warm-up iterations) of the time-varying models in the simulation study highlights the importance of the ODE solver (Supplementary S1–S4 Figs). The choice of the solver can result, in some cases, in a factor of 1, 000 difference in performance (comparing the ESS per second for Adams and trapezoidal solvers, S1 Fig). In terms of absolute error (based on RMSE), the trapezoidal solver performs best for each time-varying model. However, the relative error (based on weighted RMSE) is either higher (Brownian motion, S1 Fig) or similar (B-splines and aGPs, S2 and S4 Figs). Moreover, with B-splines and aGPs, the average performance (based on ESS per second) is increased by approximately 25%, with the ckrk solver. With aGPs, the performance also heavily depends on the choice of hyperparameters. S3 Fig shows the performance for a selection of number of basis functions and boundary factors. Across methods, 300 warm-up iterations appear sufficient for accurate model fits. For the comparison of the time-varying transmission rate models, we select the trapezoidal solver (Brownian motion) and the ckrk solver (B-splines and aGPs) and use 300 warm-up iterations.

While leading to similar results, the three approaches differ in their computational performance measured by effective sample size (ESS) per second (Fig 2E and 2F). Depending on the pre-specified knot sequence, B-splines perform up to ten times faster than GPs in our example (average of GPs implementations compared to B-splines with knot sequence 3) and up to four times faster than Brownian motion (average of Brownian motion implementations compared to B-splines with knot sequence 3). While the error in the estimates is similar between B-splines and Brownian motion (Fig 2E), the width of the 95% credible interval of the estimation of $\rho(t)$ is smaller for the Brownian motion model (Fig 2F). On the other hand, the width of the credible interval for $\rho(t)$ was similar between aGPs and B-splines (Fig 2F), and the error in the estimate is smaller for aGPs compared to B-splines (Fig 2E). For our model and simulated data, B-splines perform best in terms of statistical accuracy and computational efficiency.

Building on the best performing model specification identified with non-stratified simulated data—ascertainment by period, quasi-Poisson sampling distribution, and B-spline implementation of time-varying transmission with the aforementioned tuning parameters—we consider the fourth feature of our model: age-stratification. Again, we first validate using simulated data of an epidemic of a respiratory infection with known parameters. In this case, both the laboratory-confirmed case data and the seroprevalence data are stratified in three age groups: 0–19 years, 20–64 years and 65+, modeling the interactions between these age groups with a synthetic contact matrix. We assume that each age group can have a different ascertainment rate per period. The model correctly captures the laboratory-confirmed cases as well as the seroprevalence data, and the estimates of the age- and time-specific ascertainment rates are accurate and precise (S5 Fig). The estimation of the time-variation in the transmission rate $\rho(t)$ is in close correspondence with the true values when sufficient data are available. Larger deviations occur when the observed number of laboratory-confirmed cases is low.

In a last step, we apply the final iteration of the model, including all four essential features, to age-stratified laboratory-confirmed cases and seroprevalence data from the canton of Geneva, Switzerland, in 2020. The SARS-CoV-2 epidemic in Geneva in 2020, as in the rest of Switzerland, was characterized by a first wave in spring, low case counts in summer, followed by a severe second wave starting in the fall. Two serosurveys were conducted, once after each wave. The model is able to capture the dynamics of laboratory-confirmed cases and seroprevalence in each age group (Fig 3A). The dynamics of transmission as measured by $\rho(t)$ are very similar across age groups during the first wave, but we observed a divergence from July 2020 onwards (Fig 3B). Around this time, transmission decreased in the 65+ while remaining high all summer in the other age groups. There was then a temporary rise in transmission during fall that happened simultaneously in all age groups, but was the largest in magnitude in the 20–59 age group. Looking at ascertainment rates, we observe a clear improvement between spring and fall/winter, from 2.9% (95% CrI: 1.7–4.9%) to 23.2% (95% CrI: 18.0–31.1%) in age group 0–19, from 12.4% (95% CrI: 10.2–15.0%) to 60.7% (95% CrI: 52.4–70.9%) in age group 20–64 and from 37.8% (95% CrI: 25.1–58.1%) to 77.5% (95% CrI: 59.9–94.4%) in age group 65+.

An additional example of the B-spline based model for non-stratified data is shown in S6 Fig. Here, the model is applied to real-world SARS-CoV-2 data from the canton of Vaud, Switzerland.

    

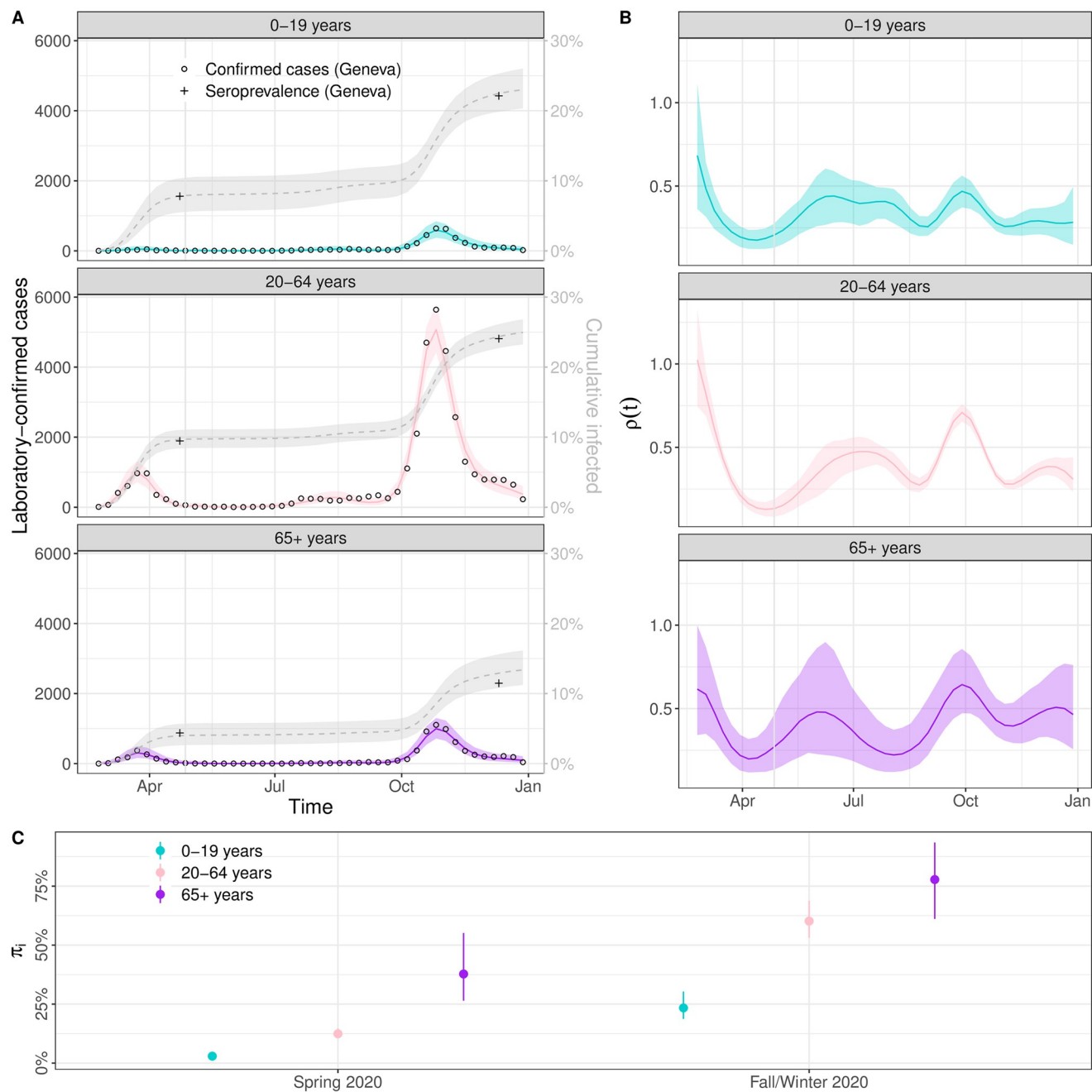

**Fig 3. Modelled SARS-CoV-2 epidemic in Geneva, Switzerland, in 2020.** (A) Posterior predictive plot for laboratory-confirmed cases (left y-axis, colored ribbon) and cumulative incidence (right y-axis, gray ribbon) per age group. Circles are weekly counts of laboratory-confirmed cases and pluses are estimates of seroprevalence at two time points. (B) Estimates of the time-varying change in transmission rate per age group using B-splines. (C) Estimates of the ascertainment rate per age group and time period.

## Discussion

Compartmental transmission models based on ODEs offer a principled and flexible way to study epidemics, but their implementation, handling, and computational efficiency for parameter inference using surveillance data can be challenging. With this study, we promote and

facilitate access to methods that allow for reliable parameter inference, full propagation of uncertainty and integration of prior knowledge in compartmental transmission models. For this purpose, we developed a Bayesian workflow aimed at studying the spread of respiratory viruses such as SARS-CoV-2 in a population over a period of time where immunity waning can be ignored, with two commonly available data sources: laboratory-confirmed case and point estimates of seroprevalence. The final iteration of the model in the workflow includes four main features deemed as essential for this task: adjustment for incomplete and differential case ascertainment across age groups, adequate sampling distribution, time-varying transmission rate and stratification by age. The exact implementation of two of these features, the sampling distribution and time-varying transmission, are the result of a benchmark and comparison of several methods using real and simulated data. We then apply this approach to real data on SARS-CoV-2 in the canton of Geneva in 2020. We also release the various model versions within the workflow as an out-of-the-box R package, where all model variations are available (https://github.com/JudithBouman2412/HETTMO or https://doi.org/10.5281/zenodo.10619448).

The Bayesian workflow, including the model comparisons we present, offers critical methodological insights into fitting compartmental transmission models to surveillance data. First, we find that the variability in laboratory-confirmed case counts for SARS-CoV-2 was best described with a quasi-Poisson distribution, which is more commonly used in ecology to describe overdispersed data [16, 17]. The correct choice of this sampling distribution is critical for both inference and short-term forecasting of epidemic dynamics and can depend on the epidemiological situation. Further research could provide additional insights into how the process noise due to stochastic transmission and superspreading in combination with the observational noise from variations in testing results in this particular distribution. Second, we add to the existing literature on time-varying transmission rates for infectious diseases, by bringing empirical evidence that forcing functions based on B-splines appear to be the most effective way to implement flexible time-varying transmission in such models, with a clear advantage over Brownian motion and aGPs [25, 27]. Choosing suitable tuning parameters can increase performance by up to a factor of 1, 000 in some cases. Our comparison includes many different specifications regarding tuning parameters, allowing us to conclude with high confidence on this open question. Third, we demonstrate that a compartmental transmission model implemented in a Bayesian framework combined with MCMC is able to handle relatively high levels of complexity, with time-varying transmission and age-stratification, thereby highlighting its potential for future methodological developments.

The application of the optimal model suggested by our workflow to the situation of the SARS-CoV-2 epidemic in the canton of Geneva, Switzerland, in 2020 highlights the practical advantages of our proposed approach. In the rather specific but critical situation of a newly-emerging respiratory pathogen circulating in a population, understanding the true level of transmission over time is of crucial importance to inform the public health response, but is generally concealed by the incomplete and unrepresentative ascertainment of cases. By combining information from laboratory-confirmed cases and serosurveys, our approach allows to estimate the ascertainment rate per age group by period bounded by seroprevalence estimates (or the emergence where seroprevalence is assumed to be null), and simultaneously to remove the effect of the ascertainment bias and determine the actual incidence of infection (with full uncertainty propagation). In the canton of Geneva, the overall ascertainment rate was estimated to be 8.6% during the first wave and 37% (95% CrI: 32–43%) during the second wave [13, 49]. These estimates from seroprevalence studies are somewhat lower than the across age group estimates from our model; (12% (95% CrI: 10–15%) and 55% (95% CrI: 47–64%)), respectively, because our second estimate includes all data since the end of the first serosurvey,

whereas Stringhini et al. (2021) calculate ascertainment for the second wave based on data between September first and December 8th only. The large differences in age-specific case ascertainment during different periods of the pandemic highlight the importance of considering age-stratified models to monitor the epidemic dynamics of viral respiratory infections. Our estimates of the time-varying change in transmission allow us to compare variation in transmission due to changes in behavior, environment and NPIs across age groups while accounting for all other aspects included in the model (such as under-ascertainment and the accumulation of natural immunity). We found a consistent reduction in the transmission rate in all age groups after the implementation of strong NPIs in spring 2020. During summer 2020, the relative transmission in 65+-year-olds was somewhat lower compared to the other age groups which could be a result of more careful social contact behavior as reported laboratory-confirmed cases numbers started to increase. At the beginning of the second wave in fall 2020, the comparatively higher transmission in 20–64 year olds compared to 0–19-year-olds is in favor of an epidemic relapse that can be attributed more to working people. The applications presented in this study rely on historical (SARS-CoV-2) data. If we can assume that the ascertainment of cases has been constant since the last available seroprevalence study, the methodology could also be used for near-real-time surveillance.

This work also has a number of limitations. First, the benchmark results are specific to our simulated data and our choice of prior distributions. We chose weakly-specific priors, only limiting the range of possible observations to plausible values [55, 56]. A prior predictive check is shown in S5 Fig. Second, our observation that the quasi-Poisson distribution best describes the noise in the ascertainment of laboratory-confirmed SARS-CoV-2 cases is specific to the data reported in the canton of Geneva. For other diseases and regions, an alternative distribution could be more appropriate. Third, it is possible that the relative performance of the three presented methods differs for distinct datasets, for instance for data collected during a longer time-period, different epidemic dynamics or a different infectious disease. The publication of our workflow in the HETTMO R package allows users to apply all presented methods and evaluate which one is the most suitable for their data. Fourth, for the approximate Gaussian process model, performance depends heavily on the choice of the hyperparameters (S3 Fig). Additionally, when applying the B-spline based method from this package, one should be aware that the performance of this method depends on the choice of the knots (Fig 2). A sufficient number of knots can be identified by subsequently increasing their number until the estimate of the transmission rate does not change any more. Fifth, our benchmark focuses on the ability of the different methods to estimate the time-varying transmission rate during the period for which data were available. We did not compare the precision for short-term forecasting, which could be done with a leave-future-out analysis. However, based on the characteristics of the methods, we would advise to use either the Brownian motion or aGP model for prediction, because for these method the variance increases with time. In contrast, B-splines are known to be at risk of error in extrapolations. Sixth, the current version of HETTMO is only useful in a limited range of situations, i.e. in a relatively short period of time following the emergence of a respiratory virus, in order to fulfill different assumptions (entirely susceptible population at the start, no vaccination, no waning of immunity and negligible changes in population sizes). The model parametrization and the way in which the incidence is coupled to the removed compartment is specific for SARS-CoV-2. However, we emphasize that the four crucial features that are the main focus of the Bayesian workflow are central for a broad range of infectious diseases. Our study provides a starting point for extensions relaxing these assumptions and in the method section we describe how the framework can be adapted to other situations and diseases.

While enormous amounts of data have been generated during the early stages of the SARS-CoV-2 pandemic, the complexity involved, with differential under-ascertainment, transmission and immunity all varying in time, creates various challenges in their interpretation. Approaches from the field of infectious disease modeling can bring invaluable insights in situations of epidemics, but require adequate, validated and efficient tools. By combining the structure of ODE-based compartmental transmission models and the power of full Bayesian inference, the presented Bayesian workflow, and its functionality in the HETTMO package provides such a tool for relatively simple situations: a newly-emerging respiratory virus spreading in a population before vaccination and immunity waning can play a role. While individual features of our workflow have been described in the literature, prior studies have not conducted a comprehensive comparison of various implementation methods to develop a complete Bayesian workflow for this specific problem. The further development of infectious disease models that can be fitted to various data sources in a Bayesian framework will promote their use for real-time monitoring, short-term forecasting, and policy making.

## Supporting information

**S1 Fig. Benchmark for Brownian motion model.** Comparison of computational performance for the Brownian motion model of the time-varying transmission rate of SARS-CoV-2 for simulated, non-stratified data for various tuning parameters: tolerance, ODE solver and number of warm-up iterations. (A) The root mean square error (RMSE) in estimating the time-variation in the transmission. (B) The RMSE weighted by the number of laboratory-confirmed cases per week. (C) The sharpness (size of the 90% confidence interval) of the time-variation in the transmission.
(TIF)

**S2 Fig. Benchmark for B-spline model.** Comparison of computational performance for the B-spline model of the time-varying transmission rate of SARS-CoV-2 for simulated, non-stratified data for various tuning parameters: tolerance, ODE solver and number of warm-up iterations. (A) The root mean squared error (RMSE) in estimating the time-variation in the transmission. (B) The RMSE weighted by the number of laboratory-confirmed cases per week. (C) The sharpness (size of the 90% confidence interval) of the time-variation in the transmission.
(TIF)

**S3 Fig. Benchmark for hyper-parameters approximate Gaussian processes model.** Analysis of the optimal number of basis functions and boundary factor for the approximate Gaussian Processes based time-varying transmission model of SARS-CoV-2 using simulated data. The number of warm-up and sampling iterations are both fixed to 300 and the trapezoidal solver is used.
(TIF)

**S4 Fig. Benchmark for approximate Gaussian processes model.** Comparison of computational performance for the approximate Gaussian processes model of the time-varying transmission rate of SARS-CoV-2 for simulated, non-stratified data for various tuning parameters: tolerance, ODE solver and number of warm-up iterations. (A) The root mean squared error (RMSE) in estimating the time-variation in the transmission. (B) The sharpness (size of the 90% confidence interval) of the time-variation in the transmission.
(TIF)

**S5 Fig. Model results for stratified simulated data.** (A) Posterior predictive plot for laboratory-confirmed cases (left y-axis, colored ribbon) and cumulative incidence (right y-axis, gray ribbon) per age group using the B-spline based age-stratified model applied to simulated data. Crosses are weekly simulated counts of laboratory-confirmed cases and pluses are simulated estimates of seroprevalence at two time points. (B) Estimates of the time-varying change in transmission rate per age group using B-splines. Crosses represent the true, simulated values. (C) Estimates of the ascertainment rate per age group and time period. Crosses represent the true, simulated values.
(TIF)

**S6 Fig. Modelled SARS-CoV-2 epidemic in Vaud, Switzerland.** (A) Posterior predictive plot for laboratory-confirmed cases (left y-axis, orange ribbon) and cumulative incidence (right y-axis, gray ribbon). Green circles are weekly counts of laboratory-confirmed cases and red triangles show monthly seroprevalence estimates from data. (B) Estimates of the time-varying change in transmission rate using B-splines. (C) Estimated ascertainment rates for first and second wave.
(TIF)

**S1 Table. Parameters for simulating unstratified data using the B-splines model.**
(TIF)

**S2 Table. Parameters adapted from the unstratified model for simulating stratified data using the B-splines.**
(TIF)

**S3 Table. Seroprevalence data from the canton of Geneva, obtained from Stringhini et al (2020) and Stringhini et al (2021) [13, 49].**
(TIF)

**S4 Table. Contact matrix indicating the average number of contacts per day of an index case (column) with individuals in three defined age groups.** This matrix is constructed using Prem et al (2021) and adjusted for the population structure of the canton of Geneva [42]. This matrix is used both for simulating stratified SARS-CoV-2 data and for analysing the data from the canton of Geneva.
(TIF)

**S1 Text. A description of the age-stratified version of the SEIR transmission model and a definition of the priors used within the Bayesian analysis.**
(PDF)

## Acknowledgments

We thank the Corona Immunitas consortium for allowing us to use their data for S6 Fig.

## Author Contributions

**Conceptualization:** Judith A. Bouman, Anthony Hauser, Samir Bhatt, Elizaveta Semenova, Andrew Gelman, Christian L. Althaus, Julien Riou.

**Formal analysis:** Judith A. Bouman, Anthony Hauser, Christian L. Althaus, Julien Riou.

**Funding acquisition:** Christian L. Althaus, Julien Riou.

**Methodology:** Judith A. Bouman, Anthony Hauser, Simon L. Grimm, Martin Wohlfender, Christian L. Althaus, Julien Riou.

**Supervision:** Christian L. Althaus, Julien Riou.

**Visualization:** Judith A. Bouman, Christian L. Althaus, Julien Riou.

**Writing – original draft:** Judith A. Bouman, Anthony Hauser, Christian L. Althaus, Julien Riou.

**Writing – review & editing:** Judith A. Bouman, Anthony Hauser, Simon L. Grimm, Martin Wohlfender, Samir Bhatt, Elizaveta Semenova, Andrew Gelman, Christian L. Althaus, Julien Riou.

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
