## [Decision Letter · Decision Letter 0]

14 Dec 2023

Dear Mrs. Bouman,

Thank you very much for submitting your manuscript "Bayesian workflow for time-varying transmission in stratified compartmental infectious disease transmission models" for consideration at PLOS Computational Biology.

As with all papers reviewed by the journal, your manuscript was reviewed by members of the editorial board and by several independent reviewers. In light of the reviews (below this email), we would like to invite the resubmission of a significantly-revised version that takes into account the reviewers' comments.

We cannot make any decision about publication until we have seen the revised manuscript and your response to the reviewers' comments. Your revised manuscript is also likely to be sent to reviewers for further evaluation.

Sincerely,

Samuel V. Scarpino

Academic Editor

PLOS Computational Biology

Virginia Pitzer

Section Editor

PLOS Computational Biology

Reviewer's Responses to Questions

**Comments to the Authors:**

Reviewer #1: # Review PLOS-CB

In their manuscript, Bouman and coauthors present a Bayesian workflow to infer time-varying parameters in stratified compartmental models. They identify four key features within this framework that need to account for i) incomplete case ascertain (a.k.a., case underreporting), ii) adequate sampling for lab-confirmed cases, iii) flexible implementation of the time-varying transmission rate, and iv) age structure. They explore different models for each and variations of the parameters thereof and validate the workflow using both synthetic and real-world data (first two COVID-19 waves in the canton of Geneva, CH). They conclude that the best-performing methods (for the disease considered and dataset quality) are quasi-poisson for sampling and BSplines for the time-varying transmission rate.

Besides providing valuable insights from the dataset they analyze (which, although regional, is novel), the article is very educational and open with the methodologies applied -- the kind of paper one would give their students to explore a new method. While I am a big fan of such papers, we have some reservations (which we list below) on the generality of the approach proposed.

## Major

1. On the generalizability of the proposed approach. While the paper, title and abstract are presented and discussed in a general way, the choice of a SEIR model (and other points listed below) are already assumptions strong enough to constrain its applicability considerably. This goes beyond the choice of parameters, which could help us translate from very similar diseases or variants thereof; it's about mechanisms. Other common respiratory viruses, as e.g. the respiratory Syncytial virus, have markedly different dynamics and thus require markedly different structures (e.g., an SIS model in cascade like that of Zheng and coauthors in JAMA Open 2021). Here, incidence and seroprevalence would have to be re-defined in terms of the new compartments and the timescales involved considered when choosing which functions to use to match the data. Some non-pharmaceutical interventions require extra compartments, and testing could also be thought of as an intervention that reduces transmissibility (as test-trace-quarantine). Vaccination and pharmaceutical interventions might not be relevant when analyzing historic 2020 COVID-19 data, but they should be when analyzing influenza. That being said, the methodology is presented in a way that is general enough to be easily adapted to account for the points I mentioned before -- please expand the methods on how to adapt the workflow to other diseases (i.e., other compartments and transition rules) and rephrase where applicable, acknowledging that the current results for real-world and synthetic data are COVID-specific. Besides, how flexible and intuitive is the R implementation of HETTMO to compartments and mechanisms? [as we are not fluent R users, we couldn't check this point ourselves]

2. On the model's structure and equations to match to observables. The SEIR engine used for simulation and inference is a strong assumption; it says that the disease that spreads is very COVID-19-like. This is not only because of the similarity with the typical disease stages of COVID (well, neglecting asymptomatic individuals), but also because of the way that seroprevalence and incidence are modelled. Defining the cumulative incidence through the compartment $R(t)$ disregards the individuals that are still in the infectious compartment but have been already tested. This works here because of the short infectious period of COVID (and the much lower estimate the authors use for $1/\\gamma$) and the smoothing induced by real-world seroprevalence studies. Furthermore, defining the modelled incidence as the difference between the recovered from one point to the other disregards the newly reported cases, which again average out with the weekly resolution of the data the authors use. While these are valid modelling choices for a COVID-19-like disease, they can be wrong assumptions for other diseases. I recommend expanding the discussions in this regard so that readers are aware of other modelling choices and possibilities.

3. On the selection of parameters. Not to be repetitive with the issue of COVID-19-like parameters (the authors do a good job discussing this in the paper), the definition of parameters in Table S6 is wrong or flipped; $\\tau$ should be the latent period, i.e., from infection to before turning infectious, and be the residence time in E. [side note, as $\\tau$ is typically thought to as with time units, defining it as a rate can be a bit unexpected for some readers and reviewers]. $\\gamma$ is typically the recovery rate, and its inverse is the days that it takes to recover (8--10)? If simulating COVID, the parameters we had in mind were $\\tau$ closer to 4--5 days, and $1/\\gamma - \\tau \\approx 4-8$ days. The definitions are inaccurate on line 78, as $\\tau$ (or its inverse) should be _the latent period_ of the disease. Also note that parameters for the contact matrix might not be available nor represent regional differences, and the mechanisms that cause the quasi-Poisson sampling to be the best might be different in other regions or when analyzing other diseases.

4. On the error function and applicability of the methodology to emerging diseases. In lines 216 and 217, it states that the model fit was performed by computing the RMSE between real and estimated values for the transmission rate $\\rho(t)$. But there can be a catch with this, which is also noted in lines 307 and 308: larger deviations occur when the number of cases is low. If that is the case, then small perturbations to the number of tests will have a drastic effect on the trends of $\\rho$, which will turn erratic. As RMSE is agnostic to the time of the measurements, deviations from these noisy measurements will weigh the same as real errors in times of high case numbers (which should be considered more reliable). For the simulated dataset, where transitions are smooth, and for the data analyzed (where high case numbers and the weekly accumulation of cases do the smoothing part) it is not a problem in general. However, how could this be accounted for in the proposed framework? Perhaps through the introduction of weights or by having a different "goal" variable?

5. On short-term forecasts and real-time surveillance data analysis. Although mentioned, the predictive power of the models is not explored. The authors mention that these models that yield results with increasing variance over time are preferred, but this could be assessed with a leave-future-out analysis of the dataset (not necessary in this case, as it is not the point of the manuscript -- but should be mentioned). However, it is not clear to us how to use this approach in real time to estimate $\\pi_t$; is it necessary to define periods of constant $\\pi$? How do we decide how many of these we need for a new dataset? How much time should we let pass before estimating $\\pi$? Discussing these questions could enlighten readers trying to analyze their datasets in the future.

## Minor

1. Figure and equations do not include stratification

2. Perhaps a demographic-inspired partition of the 18--24 would represent a general case better?

3. The author's summary could be less technical and streamlined towards the methodological/educational approach of the paper.

4. (line 10) COVID-19 pandemic instead of SARS-CoV-2 pandemic? as SARS-CoV-2 was the pathogen causing the pandemic disease.

5. About the HET en HETTMO; We are a bit skeptical about how much heterogeneity is captured here, as the core approach remains mean-field.

6. It is probably a good idea to make the color of the y-axis similar to the line color when using double y-axis (e.g.- Figure 2, the y-axis on the left side will make easier for understand that this axis is for the green line and Grey on the right y-axis for Grey line in the figure)

Reviewer #2: Review is uploaded as an attachment.

**Have the authors made all data and (if applicable) computational code underlying the findings in their manuscript fully available?**

Reviewer #1: Yes

Reviewer #2: Yes

PLOS authors have the option to publish the peer review history of their article (what does this mean?). If published, this will include your full peer review and any attached files.

Reviewer #1: No

Reviewer #2: No
---

## [Decision Letter · Decision Letter 1]

19 Mar 2024

Dear Mrs. Bouman,

Thank you very much for submitting your manuscript "Bayesian workflow for time-varying transmission in stratified compartmental infectious disease transmission models" for consideration at PLOS Computational Biology. As with all papers reviewed by the journal, your manuscript was reviewed by members of the editorial board and by several independent reviewers. The reviewers appreciated the attention to an important topic. Based on the reviews, we are likely to accept this manuscript for publication, providing that you modify the manuscript according to the review recommendations.

Reviewer 3 has made some suggestions for further clarifications to the approach and motivation for the Bayesian workflow as well as some of the supplementary figures. Reviewer 2 has suggested illustrating the approach using a second dataset on another disease, which may be beyond the scope of the current manuscript, but nevertheless the authors should consider whether this is feasible.

Finally, the Editors feel the manuscript is more appropriately designated as a "Methods" paper rather than a "Benchmarking" paper, since Benchmarking papers have additional requirements that have not been fully met (see https://journals.plos.org/ploscompbiol/article?id=10.1371/journal.pcbi.1006494). If the authors agree to this change, the journal staff should be able ensure it receives the appropriate article type designation.

Sincerely,

Samuel V. Scarpino

Academic Editor

PLOS Computational Biology

Virginia Pitzer

Section Editor

PLOS Computational Biology

Reviewer's Responses to Questions

**Comments to the Authors:**

Reviewer #1: Thank you for the consideration of our suggestions. We are satisfied with the revisions and have no further observations.

Reviewer #2: I appreciate the fact that the authors responded to comments and further explained their reasoning. However, I still think the authors should potentially perform a second analysis using their R package on a different set of COVID-19 data to further show the robustness of their analytical approach.

Reviewer #3: The authors have provided a careful example of a Bayesian workflow for a complex infectious disease modeling task. That is to say, they have presented a fitted model for a particular data set along with an iterative search of models, some of which may be applicable to related models for other data sets. I think this is a great way to go about presenting modeling results.

The authors seem to have adequately addressed the concerns raised by the initial reviewers. My own suggestion would be for them to further explain what they mean by a Bayesian workflow, as the terms seems relatively uncommon at present. What are they benefits to this approach in general and can they explain some implications of their findings in creating the workflow, such as the relative efficiency of the different ODE solvers and time-varying functions?

A minor correction the authors should make is to provide explanations for the weighed error column in S1 Fig and S2 Fig.

**Have the authors made all data and (if applicable) computational code underlying the findings in their manuscript fully available?**

Reviewer #1: Yes

Reviewer #2: Yes

Reviewer #3: Yes

PLOS authors have the option to publish the peer review history of their article (what does this mean?). If published, this will include your full peer review and any attached files.

Reviewer #1: No

Reviewer #2: No

Reviewer #3: No

Figure Files:

Data Requirements:

Reproducibility:

References:

---

## [Editor Report · Decision Letter 2]

12 Apr 2024

Dear Mrs. Bouman,

We are pleased to inform you that your manuscript 'Bayesian workflow for time-varying transmission in stratified compartmental infectious disease transmission models' has been provisionally accepted for publication in PLOS Computational Biology.

Best regards,

Samuel V. Scarpino

Academic Editor

PLOS Computational Biology

Virginia Pitzer

Section Editor

PLOS Computational Biology

---

## [Editor Report · Acceptance letter]

25 Apr 2024

PCOMPBIOL-D-23-01595R2 

Bayesian workflow for time-varying transmission in stratified compartmental infectious disease transmission models

Dear Dr Bouman,

I am pleased to inform you that your manuscript has been formally accepted for publication in PLOS Computational Biology. Your manuscript is now with our production department and you will be notified of the publication date in due course.

With kind regards,

Judit Kozma
